# Alteration in the Morphological and Transcriptomic Profiles of *Acinetobacter baumannii* after Exposure to Colistin

**DOI:** 10.3390/microorganisms12081644

**Published:** 2024-08-11

**Authors:** Eun-Jeong Yoon, Jun Won Mo, Jee-woong Kim, Min Chul Jeong, Jung Sik Yoo

**Affiliations:** 1Division of Antimicrobial Resistance Research, Korea National Institute of Health, Korea Disease Control and Prevention Agency, Cheongju-si 28159, Republic of Korea; junmomo2000@gmail.com (J.W.M.); jkjk309@korea.kr (M.C.J.); jungsiku@korea.kr (J.S.Y.); 2Division of Research Support, Korea National Institute of Health, Korea Disease Control and Prevention Agency, Cheongju-si 28159, Republic of Korea; wngsway@korea.kr

**Keywords:** *Acinetobacter baumannii*, colistin exposure, transcriptomics

## Abstract

*Acinetobacter baumannii* is often highly resistant to multiple antimicrobials, posing a risk of treatment failure, and colistin is a “last resort” for treatment of the bacterial infection. However, colistin resistance is easily developed when the bacteria are exposed to the drug, and a comprehensive analysis of colistin-mediated changes in colistin-susceptible and -resistant *A. baumannii* is needed. In this study, using an isogenic pair of colistin-susceptible and -resistant *A. baumannii* isolates, alterations in morphologic and transcriptomic characteristics associated with colistin resistance were revealed. Whole-genome sequencing showed that the resistant isolate harbored a PmrB_L208F_ mutation conferring colistin resistance, and all other single-nucleotide alterations were located in intergenic regions. Using scanning electron microscopy, it was determined that the colistin-resistant mutant had a shorter cell length than the parental isolate, and filamented cells were found when both isolates were exposed to the inhibitory concentration of colistin. When the isolates were treated with inhibitory concentrations of colistin, more than 80% of the genes were upregulated, including genes associated with antioxidative stress response pathways. The results elucidate the morphological difference between the colistin-susceptible and -resistant isolates and different colistin-mediated responses in *A. baumannii* isolates depending on their susceptibility to this drug.

## 1. Introduction

*Acinetobacter baumannii* is a clinically important Gram-negative human pathogen that causes various infections, mainly in immunocompromised patients hospitalized in intensive care units [1]. This bacterial species is successful in nosocomial environments due to its ability to persist on dry surfaces, to resist disinfectants, and to acquire genes for antimicrobial resistance. By displaying high rates of resistance to multiple drugs, including carbapenems, *A. baumannii* has become one of the largest threats to patients with bacterial infections by critically limiting their treatment options [1]. The emergence of multidrug-resistant *A. baumannii* led to the resurgence of old drugs [2,3].

Polymyxins, including colistin, are cyclic lipopeptide antimicrobials produced by the soil bacterium *Paenibacillus polymyxa* [4]. Although colistin showed significant activity against various Gram-negative pathogens, the use of the drug was stopped in the 1970s due to its nephro- and neurotoxicity [5], until it was recalled for use against multidrug-resistant bacteria. Amphiphilic colistin electrostatically interacts with polyanionic lipopolysaccharides (LPSs) in the bacterial membrane, displacing the divalent cations from the lipid A component of LPSs and thereby disrupting the outer membrane of Gram-negative bacteria [3].

Two main mechanisms of colistin resistance by alterations in the chromosomal gene have been characterized in *A. baumannii* through: (i) inactivation of the lipid A biosynthetic pathway-associated genes, resulting in complete loss of the LPSs [6], and (ii) overproduction of enzymes modifying the lipid A, such as phosphoethanolamine, decreasing the net charge of the outer membrane by adding a cationic group [7]. Since the total loss of LPSs imposes a high biological cost upon the bacterial host [8], the former mechanism is rarely identified, but the latter one is common in clinical settings. Chromosomal mutations in the *pmrA* or *pmrB* genes, encoding the PmrAB two-component system, cause overproduction of PmrC phosphoethanolamine transferase, which is responsible for modifying lipid A and conferring colistin resistance upon *A. baumannii*. Some studies have reported amino acid alterations in PmrC, which also confers colistin resistance, but the actual role of the alterations was not clearly demonstrated [7,9]. Supplementarily, possible involvement of the multidrug efflux pump overproduction, such as EmrAB, was proposed to be associated with colistin resistance [10]. In addition, through a horizontal gene transfer, a plasmid-borne gene for colistin resistance, called *m*obilized *c*olistin *r*esistance (*mcr*), has been reported in *Escherichia coli* isolates in China [11]. The *mcr-1* gene-carrying plasmid could be introduced into *A. baumannii* on a bench, resulting in reduced susceptibility to colistin [12]; however, no studies have reported *A. baumannii* clinical isolates harboring the gene [13].

In this study, using an isogenic pair of colistin-susceptible and colistin-resistant isolates, new insights into the unique signature of colistin resistance and, especially, the effects of colistin treatment on both isogenic isolates were obtained. The physical, genomic, transcriptomic, and colistin-induced transcriptomic changes were evaluated in these isolates when exposed to subinhibitory or inhibitory concentrations of colistin.

## 2. Materials and Methods

### 2.1. Bacterial Isolates

A clinical *A. baumannii* isolate Z0317AB0082 was recovered from a transtracheal aspiration sample of a patient hospitalized in 2017 in South Korea. From the parental isolate, the isogenic mutant Z0317AB0082-R was obtained on Szybalski gradient media with 0–100 µg/mL colistin. Briefly, Szybalski gradient agar was prepared in a range between 0 and 100 µg/mL of colistin (Sigma-Aldrich, Saint Louis, MO, USA), and an overnight culture of the parental isolate was applied onto the gradient agar and cultured overnight at 37 °C to recover colonies of spontaneous mutants.

### 2.2. Antimicrobial Susceptibility Testing

Disk diffusion testing for drugs other than colistin was conducted using antimicrobial disks (Becton Dickinson, Sparks, MD, USA) on cationic-adjusted Mueller–Hinton (MH) agar (Difco Laboratories, Detroit, MI, USA) according to Clinical and Laboratory Standards Institute (CLSI) guidelines [14]. Minimal inhibitory concentrations (MICs) of colistin were determined by the microdilution method using cationic-adjusted MH broth according to CLSI guidelines and interpreted through the criteria [15]. Both *E. coli* ATCC 25922 and *A. baumannii* ATCC 19606 were used as quality control isolates for susceptibility testing, as well the *mcr-1*-positive *E. coli* CREC-527 strain harboring the *mcr-1* gene [16]. The colony forming units per milliliter in each well, containing a specific concentration of colistin, were determined, and for the spontaneous mutants, the heteroresistance phenotype was assessed using the manual of Sherman et al. [17].

### 2.3. Multilocus Sequence Typing

Multilocus sequence typing (MLST) was carried out according to the Oxford scheme [18]. Briefly, fragments of seven housekeeping genes (*gltA*, *gyrB*, *gdhB*, *recA*, *cpn60*, *gpi*, and *rpoD*) were amplified by PCR, and the amplicons were Sanger-sequenced to determine the allele numbers. The sequence type of the isolate was determined using the PubMLST database (http://pubmlst.org, accessed on 27 March 2023).

### 2.4. Determination of Colistin-Resistance-Associated Gene Sequences

The *pmrCAB* operon and the genes for the lipid A biosynthesis system were amplified by PCR, and the nucleic acid sequences were determined by using Sanger sequencing. The primers used for the analysis were derived from Lesho et al. [19] for the *pmrC*, *pmrA*, and *pmrB* genes and Chen et al. [20] for the *lpxA*, *lpxC*, and *lpxD* genes.

### 2.5. RNA Isolation and Reverse Transcription Quantitative PCR (RT-qPCR)

Total RNA from both *A. baumannii* isolates was extracted from cultures in the exponential growth phase in MH broth (optical density at 600 nm = 0.8) using the RNeasy Mini Kit (Qiagen Inc., Hilden, Germany). The expression of the *pmrC*, *pmrA*, and *pmrB* genes was quantified by RT-qPCR [19] and normalized to that of the *rpoB* gene [21]. RT-qPCR reactions were performed on an Applied Biosystems™ 7500 Fast Dx Real-Time PCR instrument (Applied Biosystems, Waltham, MA, USA) using the One-Step Real-Time RT-PCR Master Mix (Thermo Fisher Scientific, Waltham, MA, USA). Each experiment was performed in triplicates at least twice.

### 2.6. Whole-Genome Sequencing

Total DNAs of both the parental Z0317AB0082 and the colistin-resistant isogenic mutant Z0317AB0082-R *A. baumannii* isolates were prepared from bacterial cells grown overnight in MH broth, quality-checked using agarose gel electrophoresis, and quantified using the Qubit (Invitrogen, Waltham, MA, USA) system. The total DNA was subjected to library preparation and sequencing on PacBio single-molecule real-time (SMRT) cells (Pacific Biosciences, Menlo Park, CA, USA). The PacBio reads were retrieved from the raw data using PacBio’s SMRT Analysis software (ver. 2.3.0.) and quality control filters; a quality score of 85 or greater and a length of 13 kbp or greater were used to trim the set of reads to a manageable size of 300 Mbp. These reads were combined with available Illumina HiSeq data using Prooveread (ver. 2.14) with default parameters. Using Bowtie 2 (ver. 2.3.5.), mapped reads were obtained from the Illumina reads and the resulting file was applied to the variant calling software FreeBayes (ver. 1.3.4).

### 2.7. Comparative Transcriptomics

Using TRIzol (Invitrogen), total RNAs were extracted from the parental *A. baumannii* Z0317AB0082 and its isogenic mutant Z0317AB0082-R isolates that had been induced for 30 min with either 2 µg/mL for both isolates or 64 µg/mL colistin for the isogenic mutant Z0317AB0082-R. Ribosomal RNA was eliminated using the NEB Nextra RNA Depletion Kit (New England Biolabs, Ipswich, MA, USA). The RNA integrity number was evaluated using the BioAnalyzer 2100 RNA Pico Chip (Agilent Technologies, Santa Clara, CA, USA), and the concentration was evaluated using Nanodrop. The RNA library was prepared with 2 µg total RNA using a TruSeq RNA Library Prep Kit (Illumina), and its quality was controlled using TapeStation HS D5000 Screening Tape (Agilent Technologies) and LightCycler qPCR. The sequencing data were 2 Gb per sample. Using the read-trimmed raw reads from Illumina HiSeq with a minimum length of 50 bp, Bowtie 2 (ver. 2.3.5.) was employed for mapping, and the value of fragments per kilobase of transcript per million mapped reads was used to normalize the sequencing depth and transcript length. Differentially expressed gene analysis was conducted using the DEGseq package of R.

### 2.8. Scanning Electron Microscopy (SEM)

For SEM, bacteria were prefixed in 2% paraformaldehyde (Sigma-Aldrich) and 2.5% glutaraldehyde in 0.1 M phosphate buffer (pH 7.4) to prevent autolysis. To minimize chemical reactions between pre- and postfixation states of the sample, the specimens were washed thrice with the same buffer as that in the fixative solution and postfixed with 1% osmium tetroxide. After washing thrice with deionized water, ascending concentrations of ethanol (50%, 70%, 80%, 90%, 95%, and 100%) were used for dehydration, which was later substituted with hexamethyldisilazane (Sigma-Aldrich). The specimens were ion-coated with platinum and observed under a field-emission scanning electron microscope (JSM-7800F, JEOL, Japan) at an acceleration voltage of 15 kV and a working distance of 5–12 mm.

### 2.9. Statistical Analysis

Data are presented as means ± standard errors (SEs) for each group. Differences in quantitative measurement were assessed by Student’s *t* test via Microsoft Excel, and a *p* value of <0.05 was considered significant.

## 3. Results

### 3.1. The A. baumannii Clinical Isolate and the Spontaneous Mutant

The Z0317AB0082 *A. baumannii* clinical isolate, isolated from the transtracheal specimen of a patient, belonged to the ST191 clone with allele numbers of *gltA*-*gyrB*-*gdhB*-*recA*-*cpn60*-*gpi*-*rpoD* of 1-3-3-2-2-94-3, respectively. Acquired resistance phenotypes of the isolate were tested using antimicrobial agents spanning eight categories (Table 1). The isolate was resistant to all extended-spectrum cephalosporins, i.e., cefotaxime, ceftriaxone, cefoxitin, ceftazidime, and cefepime; all carbapenems, i.e., ertapenem, imipenem, meropenem, and doripenem; penicillins plus beta-lactamase inhibitors, i.e., ampicillin–sulbactam; and fuloroquinolones, i.e., ciprofloxacin. For aminoglycosides, the isolate was nonsusceptible to gentamicin but remained susceptible to tobramycin and amikacin. For tetracyclines, the isolate was nonsusceptible to tetracycline but susceptible to minocycline. Since the isolate remained susceptible to two antimicrobial categories, folate pathway inhibitors including trimethoprim–sulfamethoxazole and polymyxins including colistin, it was categorized as extensively drug-resistant according to Magiorakos’s definition [22].

The colistin MIC for the colistin-susceptible parental isolate Z0317AB0082 was determined to be 2 µg/mL by the broth microdilution method, and that of the colistin-resistant isogenic mutant Z0317AB0082-R was 64 µg/mL. Both the parental isolate and the colistin-resistant isogenic mutant had an identical antibiogram except for colistin (Table 1). The *pmrC*, *pmrA*, and *pmrB* genes of the *pmrCAB* operon and the *lpxA*, *lpxC*, and *lpxD* genes of the lipid A biosynthesis pathway were amplified by PCR, sequenced, and aligned against the nucleic acid sequences of the parental isolate. For all tested colonies, the only alteration noted was *pmrB*_C625T_, resulting in the PmrB_L208F_ alteration. Only one of the mutants was chosen for further evaluation.

### 3.2. Genomic Features of the Z0317AB0082 Parental Isolate and the Colistin-Resistant Isogenic Mutant

The genome of the parental *A. baumannii* Z0317AB0082 isolate comprised a 4,012,716 bp long chromosome with two cryptic plasmids 100,260 bp and 8731 bp long. The isolate carried the intrinsic *bla*_OXA-66_ gene, encoding oxacillinase, and the *bla*_ADC-25_-like gene, encoding cephalosporinase, in the chromosome, without any upstream insertion sequences. Another intrinsic *ant(3″)-II* gene, encoding the aminoglycoside-modifying enzyme [24], was also found in the chromosome. As an acquired resistance gene, two copies of the *bla*_OXA-23_ gene were found 2968 kbp apart in the chromosome, conferring high-level resistance to the carbapenems. In the quinolone-resistance-determining region, the mutations S83L in DNA gyrase subunit A and S80L in topoisomerase IV subunit A were identified, corresponding to the resistance phenotype to ciprofloxacin.

Compared with the nucleic acid sequence of the parental Z0317AB0082 isolate, the Z0317AB0082-R isolate harbored 13 single-nucleotide polymorphisms (SNPs) and one double T insertion (Table 2). Of the 13 SNPs, 9 were in the middle of six coding sequences (CDSs), and all except *pmrB*_C625T_ were in the CDSs of transferases of insertion sequences. The other SNPs, including one in a plasmid, were in intergenic regions including the insertion case (Table 2). 

### 3.3. Morphological Alterations Due to the PmrB Mutation

The parental and colistin-resistant isogenic mutant isolates were either untreated or treated for 30 min with subinhibitory or inhibitory concentrations of colistin, following which they were imaged using SEM (Figure 1). The untreated colistin-resistant *A. baumannii* isolate, with an average length of 0.976 μm, was 10% shorter than the untreated parental isolate (1.082 μm long). Both displayed a smooth and intact surface (Figure 1A,C). All colistin-treated groups, regardless of the concentration, showed shrunken cells and wrinkled cell surfaces along with cell debris (Figure 1B,D,E). The parental isolate treated with an inhibitory concentration of colistin 2 µg/mL presented a wrinkled cell surface along with bacterial filamentation. Cells were shrunken, and cell debris was observed (Figure 1B). The colistin-resistant isogenic isolate treated with the inhibitory concentration of colistin 64 µg/mL displayed a morphology similar to the correspondingly treated parental isolate: collocation of the filamented bacterial cells and shrunken cells with cell debris (Figure 1E), while that treated with 2 µg/mL of colistin had only shrunken cells (Figure 1D).

### 3.4. Effects of Colistin on the Transcriptomic Profiles of the Colistin-Susceptible and -Resistant Isolates

By RT-qPCR, the level of expression of *pmrC*, *pmrA*, and *pmrB* genes in the colistin-resistant isogenic mutant were determined to be 1.3-, 1.7-, and 1.1-fold higher, respectively, than those in the parental isolate (Figure 2) and none of the differences were statistically significant. These three genes were overexpressed when cells were induced by inhibitory concentrations of colistin. A 30 min treatment with 2 μg/mL of colistin induced a 1.5-, 2.3-, and 2.3-fold increase in the expression of *pmrC*, *pmrA*, and *pmrB* in the parental isolate; however, no such overexpression was detected in the colistin-resistant isolate, for which 2 µg/mL was a subinhibitory concentration. At an inhibitory concentration of 64 mg/L of colistin, the colistin-resistant isolate showed a 2.1-, 1.7-, and 2.2-fold increase in expression of *pmrC*, *pmrA*, and *pmrB*, respectively.

The transcriptional status of 3731 genes was determined, and fragments per kilobase of transcript per million mapped reads values were calculated and normalized using the average expression level of six 16S rDNAs. In parental *A. baumannii* treated with inhibitory concentrations of colistin, a total of 2435 genes (over 80% of the total genes) were upregulated (Table 3), and they were associated with various functions, including responses to different stresses, such as oxidation, ultraviolet radiation, chemicals, and heavy metals. In the colistin-resistant isolate, 2629 genes were upregulated upon exposure to inhibitory concentrations of colistin, while 2021 were upregulated upon exposure to subinhibitory concentrations (Table 3). Among these, 199 genes exhibited gradual upregulation as the colistin concentration increased (Figure 3). The relative transcriptional level of each gene is indicated in Appendix A.

## 4. Discussion

Colistin is the last resort against carbapenem-resistant *A. baumannii*. Its usage in clinical settings is ever-growing; thus, understanding colistin resistance and the impact of colistin exposure is important. Here, by comparing the colistin-resistant isogenic mutant with the parental isolate, the influence of colistin on the morphology and transcriptomics of *A. baumannii* were confirmed

The mutation PmrB_L208F_, conferring colistin resistance, has been reported in *A. baumannii* clinical isolates from the period 2001 to 2008 in Spain and in the United States by the SENTRY Antimicrobial Surveillance Program [7]. The isolate was recovered from bronchial aspirates of an Italian patient being treated with colistin [25]. The leucine at 208 lies upstream of the histidine kinase domain (216–276) of the PmrB sensor kinase and its function is yet unknown. However, the previous study revealed that the mutation resulted in an overproduction of PmrC and conferred resistance to colistin by over-modifying the lipid A moiety with a MIC of 16 µg/mL. In addition, disruption of the *pmrB* coding sequence resulted in restoring the susceptibility to colistin with a MIC of 0.25 µg/mL by ceasing the over-modification [7].

In the SEM images, interesting changes in the cell length by the PmrB_L208F_ alteration can be seen. In addition, bacterial filamentation and cell elongation were observed at inhibitory concentrations of colistin, similar to the findings of Otero et al., who developed a morphology-based rapid detection method for antimicrobial resistance in Gram-negative bacteria [26]. Moreover, colistin-resistant *A. baumannii* was observed to have a shorter cell length. Filamented bacterial cells were observed when the *A. baumannii* strain were treated only by the inhibitory concentration of the colistin regardless of its susceptibility to the drug.

Previously, a transcriptomic study was conducted on a colistin-resistant isogenic mutant and its parental isolate [27]. In the current study, over five-fold overexpression was exhibited by seventeen genes in the mutant comparing with the parental isolate. Referring to the earlier results [27], six genes were overlapped: genes encoding poly-β-1,6-N-acetylglucosamine deacetylase, two putative membrane proteins, and an acetyl-CoA hydrolase and the *pmrB* and *pmrA* genes. The fold change in expression for *pmrC* was 1.8. In current study, the varying levels of expression could be due to either of the genetic alterations conferring colistin resistance, the genetic background, and possibly the different mRNA preparation methods.

Colistin resistance in clinical settings has been known to be associated mainly with the upregulation of the two-component regulatory system comprising *pmrA* and *pmrB*, which regulates the phosphoethanolamine transferase *pmrC* gene. This system was confirmed to be overexpressed after treatment with inhibitory or subinhibitory concentrations of colistin. Here, *pmrC* was moderately upregulated with a transcriptional level of 1.9. Besides the known genes associated with colistin resistance, many other genes were found to be overexpressed in the colistin-resistant isogenic isolate. Interestingly, the genes upregulated after exposure to an inhibitory concentration of colistin were different between the colistin-susceptible and the colistin-resistant isolate. 

The expression profile of colistin-associated genes has been studied in *Pseudomonas aeruginosa* [28]. Genes associated with the synthesis and modification of LPSs, oxidative stress response, and some components of the efflux pumping-out system, such as AdeABC, were found to be upregulated. Interestingly, the genes involved in antioxidative stress response pathways were overexpressed by exposure to the colistin. The upregulated genes are linked with the mechanism of action of colistin, concentrating the lethal hydroxyl radicals leading oxidative damage to DNA, lipids, and proteins upon bacteria like *A. baumannii* [29]. In the current study, the oxidative stress response regulator *soxR* was 1.3-fold upregulated in colistin-susceptible isolates treated with inhibitory concentrations of colistin, and 2.0- and 2.6-fold upregulated in the colistin-resistant isolate treated with subinhibitory and inhibitory concentrations of colistin, respectively. Similarly, the iron homeostasis-associated *fur* gene was 1.6-fold upregulated in colistin-susceptible isolates treated with inhibitory concentrations of colistin and 1.2- and 2.9-fold upregulated in colistin-resistant isolates treated with subinhibitory and inhibitory concentrations of colistin, respectively.

Since colistin resistance in Gram-negative bacteria commonly develops when the bacterial host is exposed to colistin [7,21], understanding the transcriptomic changes triggered by colistin is important. A report using the *Galleria mellonella* invertebrate animal infection model showed that colistin resistance is associated with increased virulence in vivo [30]. In addition, upregulation of the *pgaB* gene for the poly-beta-1,6-N-acetyl-D-glucosamine deacetylase was reported in colistin-resistant *A. baumannii* [27]. The protein PgaB is associated with transportation of the surface polysaccharide and an adhesin for maintenance of biofilm structural stability [31]. Correspondingly, the transcriptional level of the *pmrB* gene was three times higher in the colistin-resistant isolate than the colistin-susceptible strain in the current study. Moreover, the *pgaB* gene was 3.0-fold upregulated in colistin-susceptible isolates by colistin treatment with an inhibitory concentration and 1.5- and 1.7-fold upregulated in colistin-resistant isolates treated with subinhibitory and inhibitory concentrations of colistin, respectively. 

Infections by colistin-resistant *A. baumannii* have very limited options for treatment, and trials exploring the possibility of synergistic antimicrobial activity of colistin with other drugs were carried out. Through the transcriptomic data in this study, upregulation of several genes was observed even in subinhibitory concentrations of colistin as those in the inhibitory concentration of colistin. And other drugs with different mechanisms of action could be combined with colistin to treat the infections of colistin-resistant *A. baumannii*. A study of the in vitro synergistic efficacy of colistin combined with tigecycline, amikacin, vancomycin, rifampin, ceftazidime, imipenem, or aztreonam was carried out, and all the combinations presented synergistic activity at least for partial group of colistin-resistant *A. baumannii* clinical isolates [30]. Moreover, an in vivo study using mouse thigh and peritoneal infection models showed the combination of colistin and sulbactam eradicate better the infection-causative colistin-resistant *A. baumannii* from the lesion [31]. While many studies present the possibilities of such treatment options for the infections by colistin-resistant *A. baumannii* [32,33], clinical usage of the combinatorial therapy needs more evidence because the results from clinical trials to date are not fully convincing. A colistin combination therapy with rifampin showed inconclusive achievement of treatment for pneumoniae caused by colistin-resistant *A. baumannii* [34], and therapy with colistin combined with meropenem presented higher 28-day mortality for the patients infected with colistin-resistant *A. baumanii* than colistin monotherapy [35].

This study has several limitations. First, RNAseq results were validated using RT-qPCR for only a limited number of genes because too many genes were found to be upregulated. Second, the 13 alterations, including twelve SNPs and one insertion, other than the *pmrB*_C625T_ for the PmrBL208F mutation, were not deeply characterized for the reason that the alterations were located in the intergenic regions. Moreover, the results for the mutant with the *pmrB*_C625T_ alteration may not be generalized to mutants with other mutations. And finally, colistin-mediated virulence both in vitro and in vivo was not examined and remains open for further study. However, to the best of our knowledge, this is the first trial involving a comparative assessment of the morphological and transcriptomic profiling by colistin treatment using the isogenic pair of colistin-susceptible and -resistant isolates.

## 5. Conclusions

In this study, using a colistin-susceptible clinical *A. baumannii* isolate and its colistin-resistant isogenic mutant, colistin-mediated morphologic and transcriptomic characteristics were assessed after their exposure to colistin. The colistin-resistant mutant showed a shorter cell length than the parental isolate, and filamented cells were found when both isolates were exposed to the inhibitory concentration of colistin. Treatment with colistin led to upregulation of the genes associated with antioxidative stress response pathways, which is associated with the mechanism of antimicrobial action of colistin. In regard to the current study, future efforts are advocated to clarify the constraints of the study. 

## Figures and Tables

**Figure 1 microorganisms-12-01644-f001:**
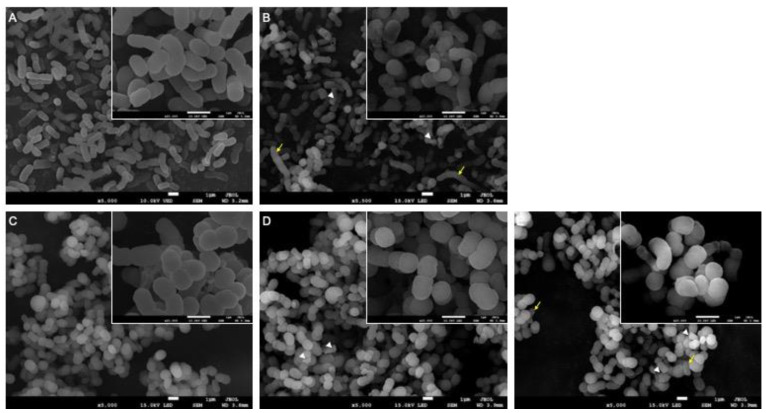
SEM micrographs of the parental Z0317AB0082 (**A**,**B**) and the colistin-resistant isogenic mutant Z0317AB0082-R (**C**–**E**) *Acinetobacter baumannii* isolates cultured in media either without (**A**,**C**) or with colistin 2 μg/mL (**B**,**D**) and 64 μg/mL (**E**). Cell rupture and shrinkage (arrow head) were observed in colistin-treated groups, and the filamentation (arrow) was observed in the groups treated by inhibitory concentrations of colistin. The images were taken at ×5000 and ×20,000 magnifications and presented with the scale bar indicating 1 μm length. For each group, triplicate samples were used, and representative images are shown.

**Figure 2 microorganisms-12-01644-f002:**
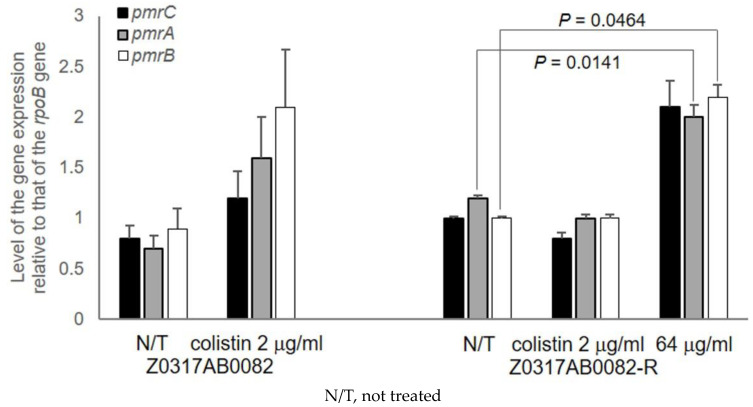
The transcriptional level of the *pmrC*, *pmrA*, and *pmrB* genes in the parental Z0317AB0082 and the colistin-resistant isogenic mutant Z0317AB0082-R isolates cultured in media either without (N/T) or with colistin 2 μg/mL and 64 μg/mL. The isolates were treated for 30 min either without or with colistin 2 μg/mL and 64 μg/mL, and the normalized transcriptional levels of *pmrC* (black), *pmrA* (gray), and *pmrB* (open) to that of the *rpoB* gene are indicated with error bars for standard error values out of biological triplicates. *p* values of <0.05 from Student’s t test are indicated.

**Figure 3 microorganisms-12-01644-f003:**
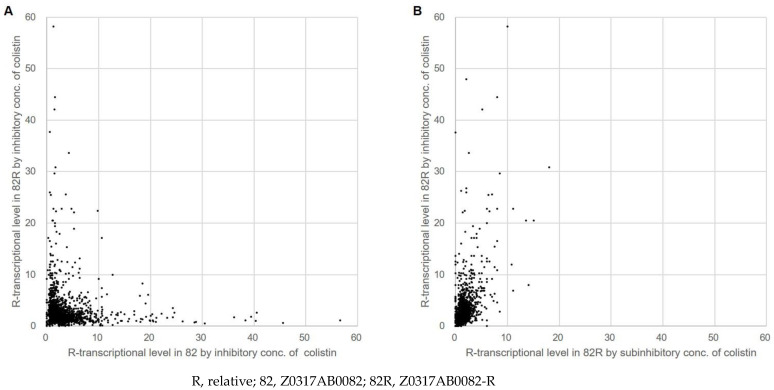
Scatter plots showing the correlations between the normalized level of transcriptome of the parental colistin-susceptible Z0317AB0082 and the colistin-resistant isogenic mutant Z0317AB0082-R isolates after treatment with inhibitory concentrations of colistin (**A**) and inhibitory or subinhibitory concentrations of colistin (**B**). The transcriptional status of a total of 3731 genes was analyzed. Each dot represents a gene. The 10 dots located over 60 for the Z0317AB0082 isolate treated with inhibitory concentrations of colistin (65.6 to 280.7), which were paired with the 0.4 to 1.9 transcriptional level in the Z0317AB0082-R isolate treated with inhibitory concentrations of colistin, were omitted from graph A for clarity.

**Table 1 microorganisms-12-01644-t001:** Antimicrobial susceptibility of the parental strain Z0317AB0082 and the colistin-resistant isogenic mutant Z0317AB0082-R.

Antimicrobial Agents	Antimicrobial Susceptibility	
	Z0317AB0082	Z0317AB0082-R
Penicillin + β-lactamase inhibitor		
Ampicillin + sulbactam	R	R
Extended-spectrum cephalosporins		
Cefoxitin	R	R
Ceftazidime	R	R
Cefepime	R	R
Carbapenems		
Ertapenem *	R	R
Imipenem	R	R
Meropenem	R	R
Doripenem	R	R
Fluoroquinolones		
Ciprofloxacin	R	R
Levofloxacin	R	R
Aminoglycosides		
Gentamicin	I	I
Tobramycin	S	S
Amikacin	S	S
Tetracyclines		
Tetracycline	R	R
Minocycline	S	S
Folate pathway inhibitor		
Trimethoprim-sulfamethoxazole	S	S
Polymyxins		
Colistin **	I	R

* The species *A. baumannii* has intrinsic resistance to ertapenem due to its carbapenem-hydrolyzing, chromosomally encoded oxacillinase represented by the OXA-51/69 variants [23] and OXA-66 in the cases of Z0317AB0082 and Z0317AB0082-R. ** The colistin MICs of Z0317AB0082 and Z0317AB0082 isolates were 2 μg/mL and 64 μg/mL, respectively.

**Table 2 microorganisms-12-01644-t002:** Single-nucleotide polymorphisms and insert/deletions found in Z0317AB0082-R compared to the parental Z0317AB0082.

No.	Type	Position	Z0317AB0082	Z0317AB0082-R
1	SNP	In a transferase CDS composing an insertion sequence	T	C
2	SNP	In a transferase CDS composing an insertion sequence	C	T
3	SNP	In the *pmrB* gene	C	T
4	SNP	In a transferase CDS composing an insertion sequence	A	T
5	SNP	In a transferase CDS composing an insertion sequence	G	A
6	SNP	In a transferase CDS composing an insertion sequence	T	C
7	SNP	In a transferase CDS composing an insertion sequence	C	G
8	SNP	In a transferase CDS composing an insertion sequence	G	A
9	SNP	In a transferase CDS composing an insertion sequence	A	C
10	SNP	Intergenic region	G	A
11	SNP	Intergenic region	T	C
12	SNP	Intergenic region	G	A
13	SNP	Intergenic region	A	G
14	Insertion	Intergenic region	GTG	GTTTG

SNP, single-nucleotide polymorphism; CDS, coding sequence; A, adenine; T, Thymine; G, guanine; C, cytosine.

**Table 3 microorganisms-12-01644-t003:** Aspects of gene expression that were changed by colistin treatment.

By Colistin Treatment	Z0317AB0082		Z0317AB0082-R			
	Inhibitory Concentration		Subinhibitory Concentration		Inhibitory Concentration	
Stay	343	11.4%	720	23.2%	311	10.0%
Downregulation	219	7.3%	369	11.9%	170	5.5%
Upregulation	2435	81.2%	2021	65.0%	2629	84.5%
Total *	2997	100%	3110	100%	3110	100%

“Stay” indicates a relative level of transcription between 0.8 and 2.0, “Downregulation” indicates a relative level of transcription below 0.8, and “Upregulation” indicates a relative level of transcription over 2.0. * Among a total of 3731 genes, the genes presenting an uncountable level of expression, either in the testing group or in the reference group, were excluded, and numbers of the total genes for transcriptomic analysis are presented.

## Data Availability

The datasets presented in this study can be found in online repositories. The name of the repository/repositories and accession number(s) can be found in the article.

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
