# Peer review of "Alteration in the Morphological and Transcriptomic Profiles of Acinetobacter baumannii after Exposure to Colistin"

_microorganisms, 2024, doi:10.3390/microorganisms12081644_

Round 1

Reviewer 1 Report

Comments and Suggestions for Authors

This study analyzed the morphological and transcriptomical changes of a colistin sensitive and a colistin resistant A. baumannii pathogen after being treated with colistin.

 It appears a well conducted study and the manuscript is generally well written.

There are some minor comments that need to be addressed:

Introduction

Lines 51-52:

«overproducing enzymatic genes decreasing in the net charge of the outer membrane by adding a cationic group, such as phosphoethanolamine, to the lipid A moiety» needs clarification.

Results

Table 2 requires footnote explaining the abbreviations.

Lines 261-263 : “In the colistin-resistant isolate, 2,629 genes were upregulated upon exposure to inhibitory concentrations of colistin, while 2021 were upregulated upon exposure to subinhibitory concentrations (Table 3). Among these, 199 genes exhibited gradual upregulation as the colistin concentration increased (Figure 3)” Authors discovered gene expression upregulation in several genes even in subinhibitory concentrations. Clinical practice has shown effectiveness of colistin treatment even in colistin resistant strains (usually as a combination with other antibiotics such as sulbactam). Could the findings of the authors explain the efficacy of colistin in colistin resistant strains? I would like the input of the authors on this.

Discussion

308-311   Though the transcriptomic analysis should have differed criteria dependent on the experimental batch, we used differed criteria of  overexpression of the gene, seventeen genes were more than five-fold overexpressed in the mutant compared with the parental isolate, of which six overlap with our findings:” meaning is unclear

Limitations section: study was performed on a PmrBL208F mutation strain. Are results generalizable to other strains with different mutations?

Comments on the Quality of English Language

a couple of phrases require clarrification see above

Reviewer 2 Report

Comments and Suggestions for Authors

The manuscript of Yoon et al. “Alteration of ...” is interesting, but needs some changes.

1.       The references are not listed in citation order

2.       Lines 33 and 43 Gram comes from the researcher's last name, so it should be capitalized

3.       Lines 78 and 88 add company and country

4.       Line 89 at first full name of bacteria should be introduced Escherichia coli, next acronym E. coli

5.       Line 105 the reference should not be in the upper index

6.       Line 151 add country

7.       Table 1 ampicillin with sulbactam is a penicillin with B-lactamase inhibitor add inhibitor

8.       Table 1 Acinetobacter is naturally resistant to some of the antibiotics in Table 1 (e.g. cefotaxime, ceftriaxone...), so there is no point in assessing susceptibility to them.

9.       Table 2 Carbapenemase is an enzyme, should be Carbapenems

10.   Table 2 the name fluoroquinolones is misspelled

11.   Table 2 Why ciprofloxacin is the only fluoroquinolone, add levofloxacin

12.   Lines 180-181 what is the purpose of the CLSI and EUCAST criteria for colistin

Reviewer 3 Report

Comments and Suggestions for Authors

Please modify your manuscript according suggested comments included in the attached manuscript.
